# Structural development and dorsoventral maturation of the medial entorhinal cortex

**Saikat Ray\*, Michael Brecht\***

Bernstein Center for Computational Neuroscience, Humboldt University of Berlin, Berlin, Germany

**Abstract** We investigated the structural development of superficial-layers of medial entorhinal cortex and parasubiculum in rats. The grid-layout and cholinergic-innervation of calbindin-positive pyramidal-cells in layer-2 emerged around birth while reelin-positive stellate-cells were scattered throughout development. Layer-3 and parasubiculum neurons had a transient calbindin-expression, which declined with age. Early postnatally, layer-2 pyramidal but not stellate-cells co-localized with doublecortin – a marker of immature neurons – suggesting delayed functional-maturation of pyramidal-cells. Three observations indicated a dorsal-to-ventral maturation of entorhinal cortex and parasubiculum: (i) calbindin-expression in layer-3 neurons decreased progressively from dorsal-to-ventral, (ii) doublecortin in layer-2 calbindin-positive-patches disappeared dorsally before ventrally, and (iii) wolframin-expression emerged earlier in dorsal than ventral parasubiculum. The early appearance of calbindin-pyramidal-grid-organization in layer-2 suggests that this pattern is instructed by genetic information rather than experience. Superficial-layer-microcircuits mature earlier in dorsal entorhinal cortex, where small spatial-scales are represented. Maturation of ventral-entorhinal-microcircuits – representing larger spatial-scales – follows later around the onset of exploratory behavior.

**\*For correspondence:** saikat.ray@bccn-berlin.de (SR); michael.brecht@bccn-berlin.de (MB)

**Competing interests:** The authors declare that no competing interests exist.

## Introduction

The representation of space in the rodent brain has been investigated in detail. The functional development of spatial response properties has also been investigated in the cortico-hippocampal system (*Ainge and Langston, 2012*; *Wills et al., 2014*), with studies suggesting the early emergence of head-directional selectivity (*Tan et al., 2015*; *Bjerknes et al., 2015*), border representation (*Bjerknes et al., 2014*) and place cell firing, but a delayed maturation of grid cell discharges (*Wills et al., 2010*; *Langston et al., 2010*).

Even though there is information on the emergence of functional spatial properties in the hippo-campal formation, remarkably little is known about the structural development of the microcircuits which bring about these properties. To understand this, we investigated the development of the architecture of the medial entorhinal cortex (MEC) and parasubiculum (PaS), two key structures in the cortico-hippocampal system.

In adult animals, layer 2 of MEC contains two types of principal cells, stellate and pyramidal cells (*Alonso and Klink, 1993*; *Germroth et al., 1989*). Stellate and pyramidal neurons are distinct in their intrinsic conductance (*Alonso and Llinás, 1989*; *Klink and Alonso, 1997*), immunoreactivity (*Varga et al., 2010*), projections (*Lingenhöhl and Finch, 1991*; *Canto and Witter, 2012*) and inhibitory inputs (*Varga et al., 2010*). Pyramidal neurons in layer 2 of MEC can be identified by calbindin-immuno-reactivity (*Varga et al., 2010*) and are clustered in patches across various mammalian species (*Fujimaru and Kosaka, 1996*; *Ray et al., 2014*; *Naumann et al., 2016*), while stellate cells can

**eLife digest** Many animals, from rats to humans, need to navigate their environments to find food or shelter. This ability relies on a kind of memory known as spatial memory, which provides a map of the outside world within the animal's brain. Specifically, cells in a part of the brain called the medial entorhinal cortex act like the grids present on a map, and are known as grid cells. Other cells in this region represent boundaries in the environment and are known as border cells. These cells and other cells connect to each other to make the spatial memory circuit.

Previous research had reported that the grid cells were not present in the very early stages of an animal's life. It was also not clear how the different cell types involved in spatial memory develop after birth. Ray and Brecht have now studied rats and found that certain characteristic structures in the circuit are present at birth. For example, cells that were most likely to become grid cells, were already laid out in a grid, indicating that this layout is instructed by genetic information rather than experience.

Ray and Brecht also found that the cells that most likely become grid cells matured later than the cells that most likely become border cells. Further analysis then revealed that the circuits in the top part of the medial entorhinal cortex, which represents nearby areas, matured earlier than those in the bottom part of this region, which represent farther areas. These findings could therefore explain why rats explore nearby areas earlier in life before going on to explore further away areas at later stages.

More work is needed to characterize other components of the neural circuits involved in spatial memory to provide a complete understanding of how these memories are formed. Future experiments could also ask if encouraging young rats to explore a wider area can cause the circuits to mature more quickly.

be identified by reelin-immuno-reactivity (*Varga et al., 2010*) and a lack of structural periodicity (*Ray et al., 2014*). In rodents, the grid-like arrangement of pyramidal cell patches is aligned to cholinergic inputs (*Ray et al., 2014*; *Naumann et al., 2016*). Functionally, about a third of all cells in layer 2 exhibit spatial tuning with grid, border, irregular and head-directional discharges being present (*Tang et al., 2014*).

Neurons in layer 3 of MEC are characterized by rather homogenous in vitro intrinsic and in vivo spatiotemporal properties (*Tang et al., 2015*). A majority of cells exhibit a lack of spatial modulation, and the remaining are mainly dominated by irregular spatial responses (*Tang et al., 2015*) with a fraction also exhibiting grid, border and head-directional responses (*Boccara et al., 2010*).

The parasubiculum is a long and narrow structure flanking the dorsal and medial extremities of MEC (*Video 1*). The superficial parasubiculum, corresponding to layer 1 of MEC is divided into large clusters, while the deeper part, corresponding to layers 2 and 3 of MEC, is rather homogenous (*Tang et al., 2016*). In terms of functional tuning of cells, a majority of the cells of PaS show spatially tuned responses, and include grid, border, head-directional and irregular spatial cells (*Boccara et al., 2010*; *Tang et al., 2016*).

Here we investigate the emergence of the periodic pyramidal-cell patch pattern in layer 2 of MEC, as well as the development of cellular markers that characterize the architecture of adult MEC and PaS. The results indicate an early emergence of pyramidal cell organization, a delayed maturation of pyramidal but not stellate cells and a dorsal-to-ventral maturation of MEC circuits.

## Results

We first investigated development of brain size and thickness of layers of the MEC (*Figure 1*) by observing rats at E18, P0, P4, P8, P12, P16, P20, P24 and adults (>P42). The majority of the brain development takes place within the first few weeks postnatally (*Figure 1a*), with the brain size increasing 1000% from 0.12 ± 0.00 g at E18 (mean ± SD; n=3) to 1.23 ± 0.07 g at P12 (n=5). Subsequently, the growth plateaus to ~25% with the brain weighing 1.71 ± 0.08 g at P24 (n=6) and having a weight of 2.11 ± 0.14 g in adults (n=9) (*Figure 1b*). The superficial layers (layers 1–3) of the MEC (*Figure 1c*) double in thickness during this early postnatal period from 243 ± 35 μm at P0 (mean ±

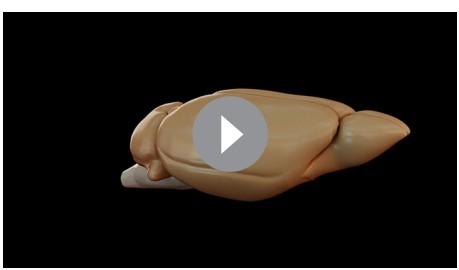

**Video 1.** Medial entorhinal cortex and parasubiculum in the rat brain. The medial entorhinal cortex and parasubiculum are situated at the posterior extremity of the rat neocortex. This schematic video illustrates the location of the medial entorhinal cortex and parasubiculum in situ, the tangential sectioning process and the layout of parasubicular patches and calbindin-patches in the medial entorhinal cortex.

SD; n=21, 4 rats) to 652 ± 50 µm at P12 (n=24, 4 rats). A similar increase is also observed in the deeper layers (layers 4–6) from 167 ± 21 µm at P0 (n=21, 4 rats) to 329 ± 54 µm at P12 (n=24, 4 rats).The overall thickness plateaus around this point to 981 ± 81 µm at P12 (n=24, 4 rats) and remains at 882 ± 78 µm in adults (n=24, 4 rats) (*Figure 1d*). Proportionally, the thickness of the layers remains similar during development, with layer 2 accounting for ∼20% and layers 3 and 5/6 each accounting for ∼30% of the MEC. Layers 1 and 4 are the thinnest at about 10% and 5% of the total thickness respectively (*Figure 1d*).

We next investigated the microcircuit organization of superficial layers of MEC. Calbindin, a calcium binding protein, is selectively expressed in layer 2 pyramidal cells (*Varga et al., 2010*; *Fujimaru and Kosaka, 1996*), which form a grid-like arrangement in adult animals (*Ray et al., 2014*). Concurrently, reelin, an extracellular matrix protein, is selectively expressed in stellate cells in layer 2 of MEC, which are scattered throughout (*Ray et al., 2014*) layer 2. To visualize the development of entorhinal microcircuits we first prepared tangential sections (see our video animation on preparing tangential sections, *Video 1*) through layer 2 of medial entorhinal cortex and stained for calbindin-immunoreactivity. From the earliest postnatal stages, calbindin+ neurons in the MEC exhibited clustering, forming patches at P0 (*Figure 2a*). The calbindin+ patches at P0 exhibited a grid-like (*Figure 2a,b*) regular arrangement (*Figure 2c*), determined by spatial autocorrelation analysis and grid scores, similar to that observed in adults (*Ray et al., 2014*; *Naumann et al., 2016*, *Figure 2d–f*). Similar preparations for visualizing stellate cells by reelin-immunoreactivity (*Figure 2—figure supplement 1*), exhibited the presence of stellate cells in early postnatal stages (*Figure 2—figure supplement 1a,b*) and a lack of periodicity (*Figure 2—figure supplement 1c*), similar to observations made in adults (*Ray et al., 2014*, *Figure 2—figure supplement 1d–f*). Calbindin+ pyramidal neurons in the MEC (*Figure 2g*) also received preferential cholinergic innervation early postnatally (*Figure 2h–i*), similar to adults (*Ray et al., 2014*; *Naumann et al., 2016*, *Figure 2j–l*).

In the parasubiculum, a transient presence of calbindin was observed with ∼15 clusters of calbindin+ neurons at P0 (*Figure 2a*) and P4 (*Figure 2g*). This expression was curtailed in older animals, with only a calbindin+ stripe persisting in adults (*Figure 2d*).

To visualize the laminar development of MEC, we stained parasagittal sections for calbindin (*Figure 3*) and reelin (*Figure 4*) immunoreactivity. Indications of calbindin+ neuronal clusters were visible prenatally at E18 (*Figure 3a*). However, the calbindin+ patches in the MEC did not exhibit clustering of their dendrites, as previously described in adults (*Ray et al., 2014*) at E18 and P0 (*Figure 3a,b*). Some dendritic clustering could be observed at P4 (*Figure 3c*), while from P8 (*Figure 3d–h*) the dendritic clustering of calbindin+ pyramidal neurons was similar to that in adults. In layer 3 of the MEC, we observed a transient presence of calbindin expression. The number of calbindin+ neurons in layer 3 declined progressively from prenatal stages to P20 (*Figure 3a–g*), where it attained adult-like levels with rarely any calbindin+ neurons in layer 3 (*Figure 3h*). Quantitatively, calbindin+ neuronal density (calbindin+ neurons per mm$^2$) decreased from 955 ± 315 (mean ± SD; count refers to n=3776 neurons in 8 rats) in P4-P8 rats to 333 ± 99 (n=2104 neurons, 8 rats) in P12-P16 rats to 141 ± 56 (n=828 neurons, 7 rats) in adults (*Figure 3i*).

Reelin was present in layer 2 from early postnatal stages (*Figure 4a*; *Figure 2—figure supplement 1a,b*), though the most prominent reelin-immunoreactive cells in the first two postnatal weeks were present in layer 1 (*Figure 4a–c*). Reelin expression increased in layer 3 of the MEC from early postnatal stages to P20 (*Figure 4a–e*), where it attained adult-like levels (*Figure 4f*). Quantitatively, reelin+ neuronal density in layer 3 increased from 729 ± 435 (n=1405 neurons, 4 rats) in P4-P8 rats to 1549 ± 115 (n=3309 neurons, 3 rats) in P12-P16 rats to 1996 ± 208 (n=5039 neurons, 3 rats) in adults.

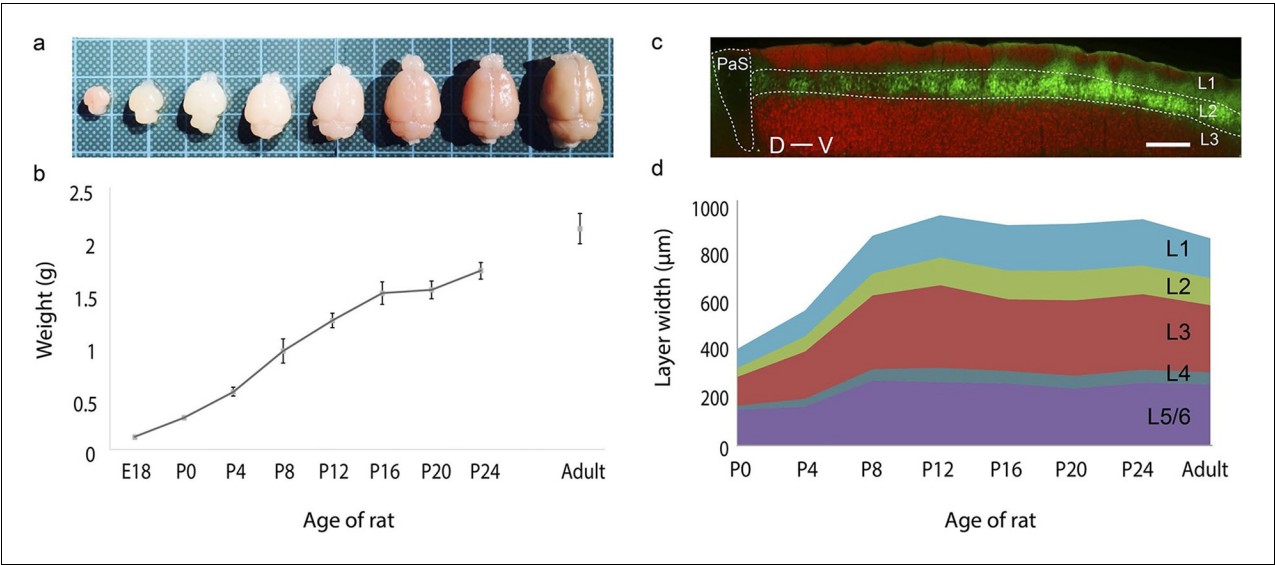

**Figure 1.** Rat brain and medial entorhinal cortex laminar development. (a) Growth in rat brain size from E18, P0, P4, P8, P12, P16, P20 to adult. Brains are overlaid on a 1 cm x 1 cm grid. (b) Mean weight (in grams) of E18 (n=3), P0 (n=6), P4 (n=5), P8 (n=5), P12 (n=5), P16 (n=5), P20 (n=5), P24 (n=6) and in adult (n=9) rat brains. Error bars indicate SD. (c) Parasagittal section double stained for calbindin-immunoreactivity (green) and Purkinje cell protein 4 immunoreactivity (pcp4; red), illustrating the superficial layers of the medial entorhinal cortex and parasubiculum. Calbindin+ neurons (green) are in layer 2, pcp4+ neurons (red) are in layer 3 MEC. (d) Development of mean layer width (in μm) of layer 1 (light-blue), layer 2 (green), layer 3 (red), layer 4 (gray-blue) and layer 5/6 (purple) from P0 to P24 and in adult rat medial entorhinal cortex. Scale bars 250 μm. PaS- Parasubiculum; L1- Layer 1; L2- Layer 2; L3- Layer 3; D- Dorsal; V-Ventral.

The following source data is available for figure 1:

**Source data 1.** Laminar widths (in μm) of the medial entorhinal cortex for P0, P4, P8, P12, P16, P20, P24 and adult rats.

Three observations indicated a dorsal-ventral developmental gradient in the superficial layers of medial entorhinal cortex and parasubiculum:

First, the transient calbindin expression in layer 3 disappeared from dorsal to ventral. Thus, most of layer 3 had calbindin+ neurons at P8 (*Figure 5a*), only the ventral half of layer 3 showed calbindin expression at P16 (*Figure 5b*), and in adults calbindin expression was largely absent from layer 3 of MEC (*Figure 5c*). This transient expression of calbindin in layer 3 followed a dorso-ventral developmental profile (*Figure 5d*). Early postnatally, in P4-P8 rats, we observed equitable densities of calbindin+ cells in dorsal, intermediate and ventral levels of MEC (n=3776 neurons, 8 rats). In contrast, around the end of the second postnatal week, in P12-P16 rats, we observed significantly lower densities (p=0.010, Mann-Whitney two tailed) in the dorsal (225 ± 96 cells / mm$^2$), as opposed to the ventral (449 ± 161 cells / mm$^2$) MEC (n=2104 neurons, 8 rats). In adults (n=828 neurons, 7 rats), calbindin+ neurons were largely absent in layer 3, but among the remaining population the density waxed from dorsal to intermediate and ventral MEC. The development of reelin expression in layer 3 neurons on the other hand (*Figure 5—figure supplement 1a–c*) occurred in equitable proportions in dorsal, intermediate and ventral levels of MEC (*Figure 5—figure supplement 1d*) with increasing age.

Second, layer 2 calbindin+ patches in the MEC also exhibited a dorsal-to-ventral maturation profile. The calbindin+ patches (*Figure 6a*) co-localized with doublecortin (*Figure 6b*), a well-established marker for immature neurons (*Brown et al., 2003*) throughout layer 2 at P8 (*Figure 6c,d*). At P16, the dorsal calbindin+ patches (*Figure 6e,g*) did not express doublecortin (*Figure 6f,g*), while ventral calbindin+ patches still co-localized with doublecortin (*Figure 6h*). In adults, calbindin+ patches (*Figure 6i*) did not exhibit doublecortin (*Figure 6j*) in either dorsal (*Figure 6k*) or ventral (*Figure 6l*) parts. A similar dorsal-to-ventral development gradient was evident in the PaS, with doublecortin being present throughout the PaS in P8 (*Figure 6b*), only in the ventral part in P16 (*Figure 6f*) and not present in adults (*Figure 6j*). To quantify the overlap between calbindin and

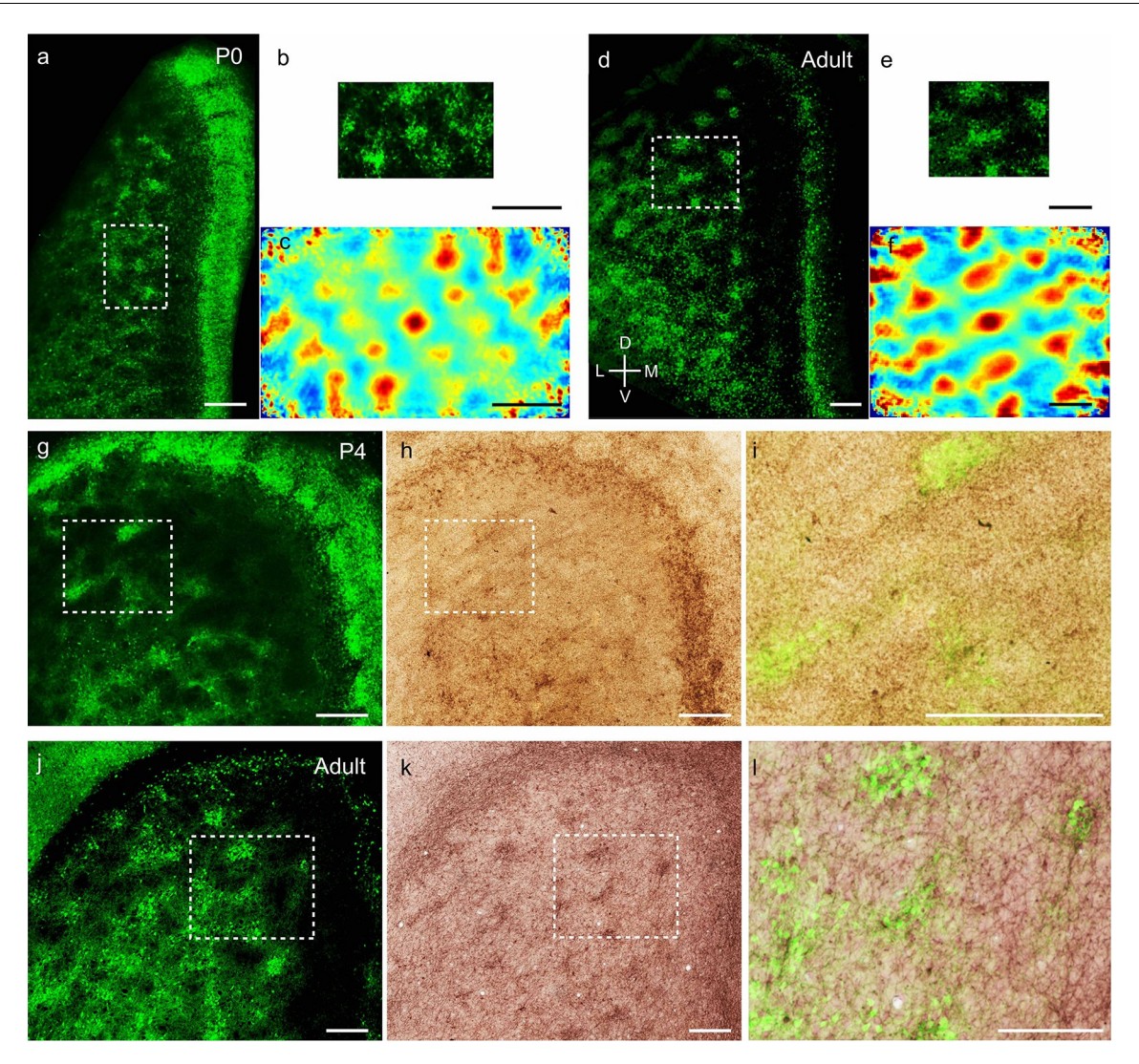

**Figure 2.** Adult-like grid layout and cholinergic innervation of calbindin+ pyramidal neurons in layer 2 of MEC at early postnatal stages. (a) Tangential sections of the MEC processed for calbindin-immunoreactivity (green). Patches of calbindin+ neurons are evident already in the MEC, while the parasubicular patches at the right extremity also show calbindin-immunoreactivity in P0 rats. (b) Inset from (a), rotated 90 degrees clockwise, for presentation. (c) Two-dimensional spatial autocorrelation of the MEC region shown in (b) showing a periodic spatial organization of calbindin+ patches. The grid score is 0.59. (d) as (a) for adult animals. (e) Inset from (d). (f) Two-dimensional spatial autocorrelation of the MEC region shown in (e) showing a periodic spatial organization of calbindin+ patches. The grid score is 1.18. (g) Tangential section in a P4 animal processed for calbindin-immunoreactivity (green). Also note the calbindin-immunoreactive parasubicular patches present in a P4 rat. (h) Section from (g) co-stained for acetylcholinesterase activity (brown). (i) Overlay of inset regions from (g) and (h) shows overlap between calbindin and acetylcholinesterase in MEC in P4 rats. (j–l) as (g–i) for adult animals. (d–f, j–l) modified from *Ray et al. (2014)*. Colour scale of spatial autocorrelation, -0.5 (blue) through 0 (green) to 0.5 (red). Scale bars 250 µm. D- Dorsal; V- Ventral; M- Medial; L- Lateral. Orientation in (d) applies to all sections apart from (b), where it's rotated 90 degrees clockwise.

Figure 2, panels d-f and j-l are adapted from Ray S, Naumann R, Burgalossi A, Tang Q, Schmidt H, Brecht M. 2014. Grid-layout and theta-modulation of layer 2 pyramidal neurons in medial entorhinal cortex. Science 343:891–896. doi:10.1126/science.1243028. Reprinted with permission from AAAS (© copyright The American Association for the Advancement of Science, 2014).

The following figure supplement is available for figure 2:

**Figure supplement 1.** Adult-like scattered distribution of reelin+ stellate cells in early postnatal stages.

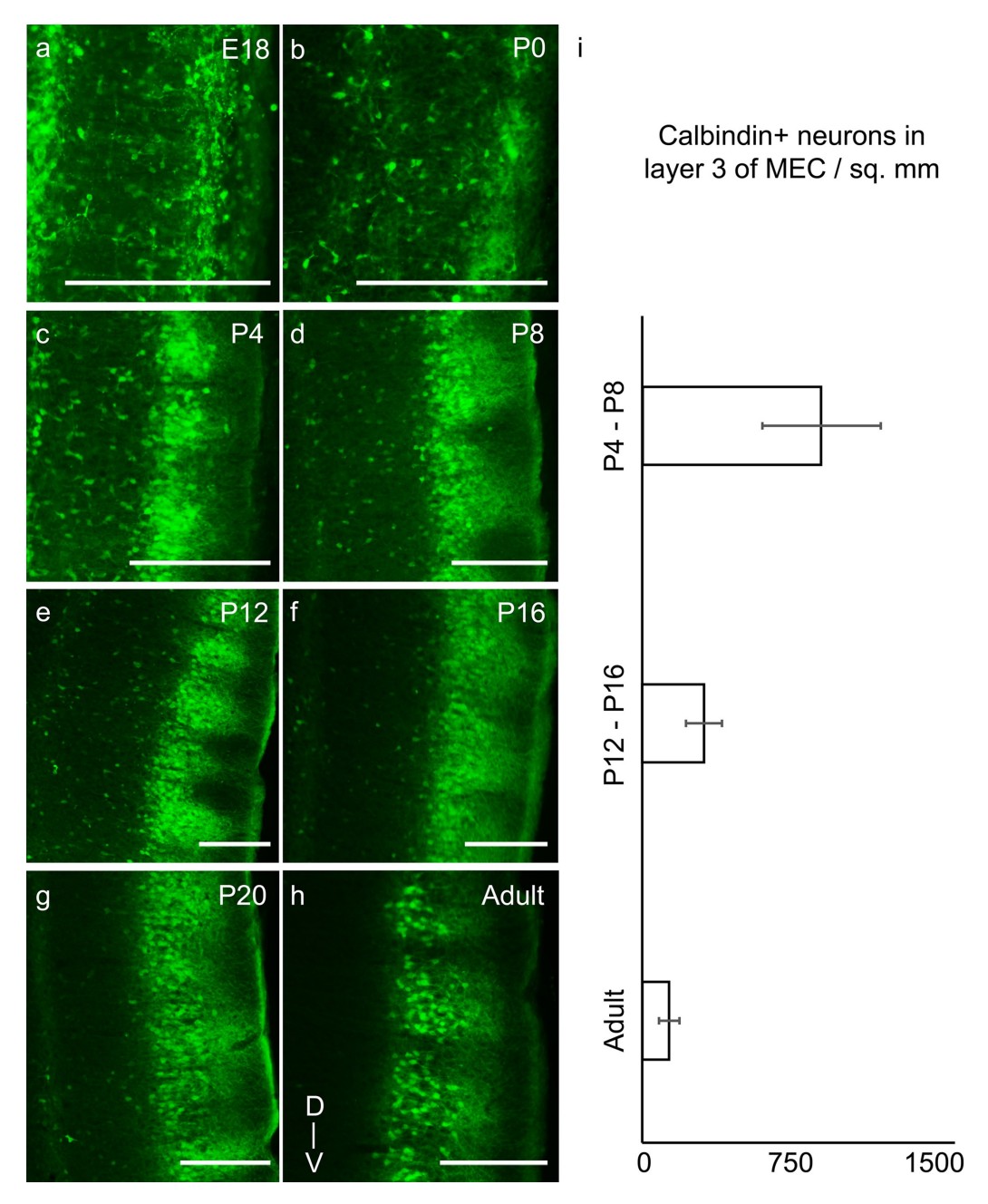

**Figure 3.** Transient presence of calbindin+ neurons in layer 3 of MEC in early postnatal stages reduces progressively to adult-like state by third postnatal week. Parasaggital sections of the MEC processed for calbindin-immunoreactivity (green). The sections show clustering of calbindin+ pyramidal cells in layer 2 and a transient presence of calbindin+ neurons in layer 3, which decrease with age in (**a**) E18 rat. (**b**) P0 rat. (**c**) P4 rat. (**d**) P8 rat. (**e**) P12 rat. (**f**) P16 rat. (**g**) P20 rat. (**h**) Adult rat. (**i**) Decreasing density of calbindin+ neurons in layer 3 of MEC from P4-P8 (n=3776 neurons, 8 rats); to P12-P16 (n=2104 neurons, 8 rats) to adults (n=828 neurons, 7 rats). Error bars denote SD. Scale bars 250 μm. D- Dorsal; V- Ventral. Orientation in (**h**) applies to all sections.

The following source data is available for figure 3:

**Source data 1.** Calbindin+ neurons counted and areas (in μm$^2$) in layer 3 for determining calbindin+ neuronal density in layer 3 in P4-P8, P12-P16 and adult rats.

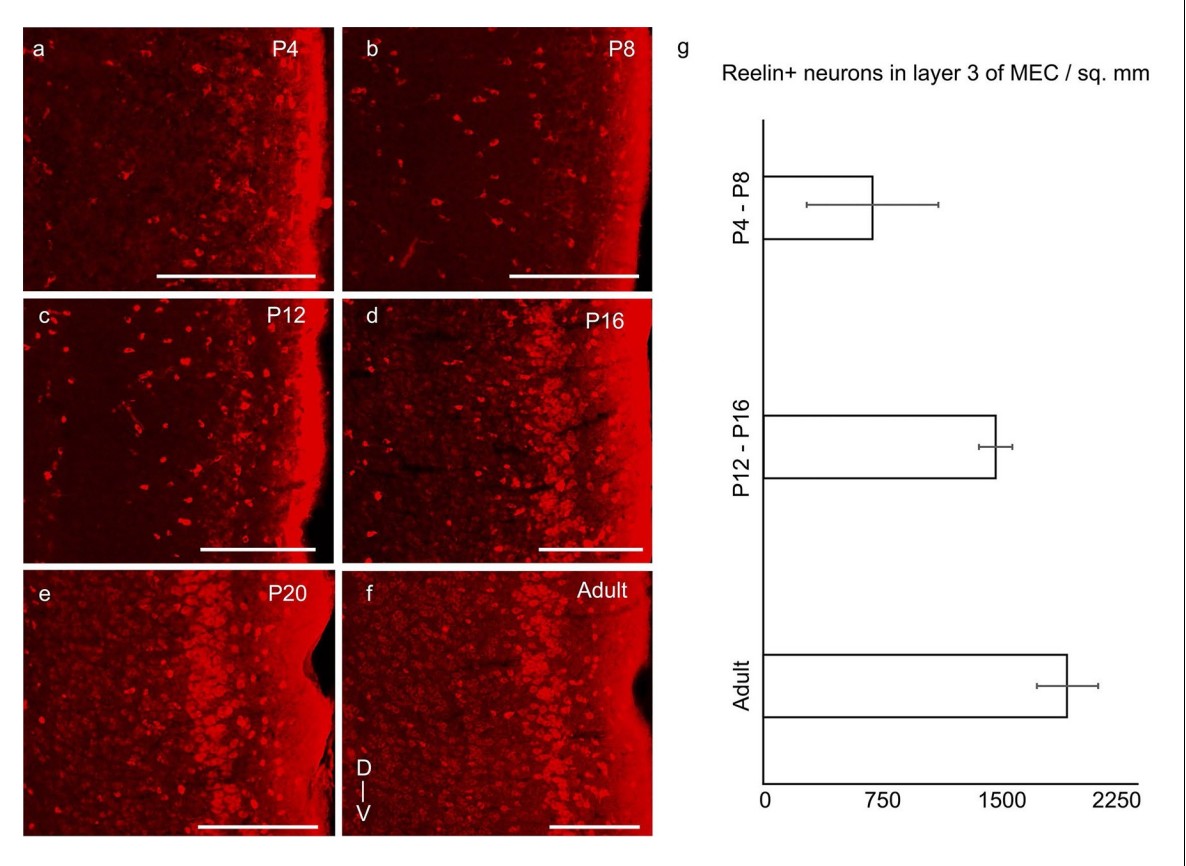

**Figure 4.** Increase of reelin expression in layer 3 neurons of MEC through development. Parasaggital sections of the MEC processed for reelin-immunoreactivity (red). The sections show reelin+ stellate cells in layer 2 and an increasing reelin expression in layer 3 neurons with development in (**a**) P4 rat. (**b**) P8 rat. (**c**) P12 rat. (**d**) P16 rat. (**e**) P20 rat. (**f**) Adult rat. (**g**) Increasing density of reelin+ neurons in layer 3 of MEC from P4-P8 (n=1405 neurons, 4 rats); to P12-P16 (n=3309 neurons, 3 rats) to adults (n=5039 neurons, 3 rats). Error bars denote SD. Scale bars 250 μm. D- Dorsal; V- Ventral. Orientation in (**f**) applies to all sections.

The following source data is available for figure 4:

**Source data 1.** Reelin+ neurons counted and areas (in μm$^2$) in layer 3for determining reelin+ neuronal density in layer 3 in P4-P8, P12-P16 and adult rats.

doublecortin we performed spatial cross-correlations (**Figure 6m**). P8-P12 rats exhibited a high degree of overlap between calbindin and doublecortin in both dorsal (0.74 ± 0.05; mean ± SD, Pearson's cross-correlation coefficient) and ventral (0.61 ± 0.10) parts (n=9 regions, 5 rats). In P16-P20 rats (n=16 regions, 8 rats), the dorsal regions showed low correlations (0.14 ± 0.17), while the ventral regions still showed significantly higher overlap (0.60 ± 0.07; p=0.0008, Mann-Whitney two tailed). In adults (n=7 regions, 4 rats), both dorsal (0.19 ± 0.07) and ventral (0.20 ± 0.07) regions had low overlap. The difference in the Pearson's cross correlation coefficient between overlapping regions (dorsal and ventral in P8-P12; ventral in P16-P20) and non-overlapping regions (dorsal in P16-P20; dorsal and ventral in adults) was significant at p=0.000001 (Mann-Whitney two tailed).

A closer analysis of the co-localization of the immature neuronal marker doublecortin with calbindin+ pyramidal cells and reelin+ stellate cells (**Figure 7a–c**) revealed doublecortin to be mostly co-localized with calbindin+ rather than reelin+ neurons (**Figure 7d**). Spatial cross-correlations between doublecortin and either calbindin or reelin (**Figure 7e**; n=8 rats from ages P8 - P20) from triple-immunostained calbindin, reelin and doublecortin regions of layer 2 of the MEC revealed a greater overlap of doublecortin with calbindin (0.54 ± 0.10) than with reelin (0.08 ± 0.13). This difference in the Pearson's cross correlation coefficient was significant at p=0.0009 (Mann-Whitney two tailed).

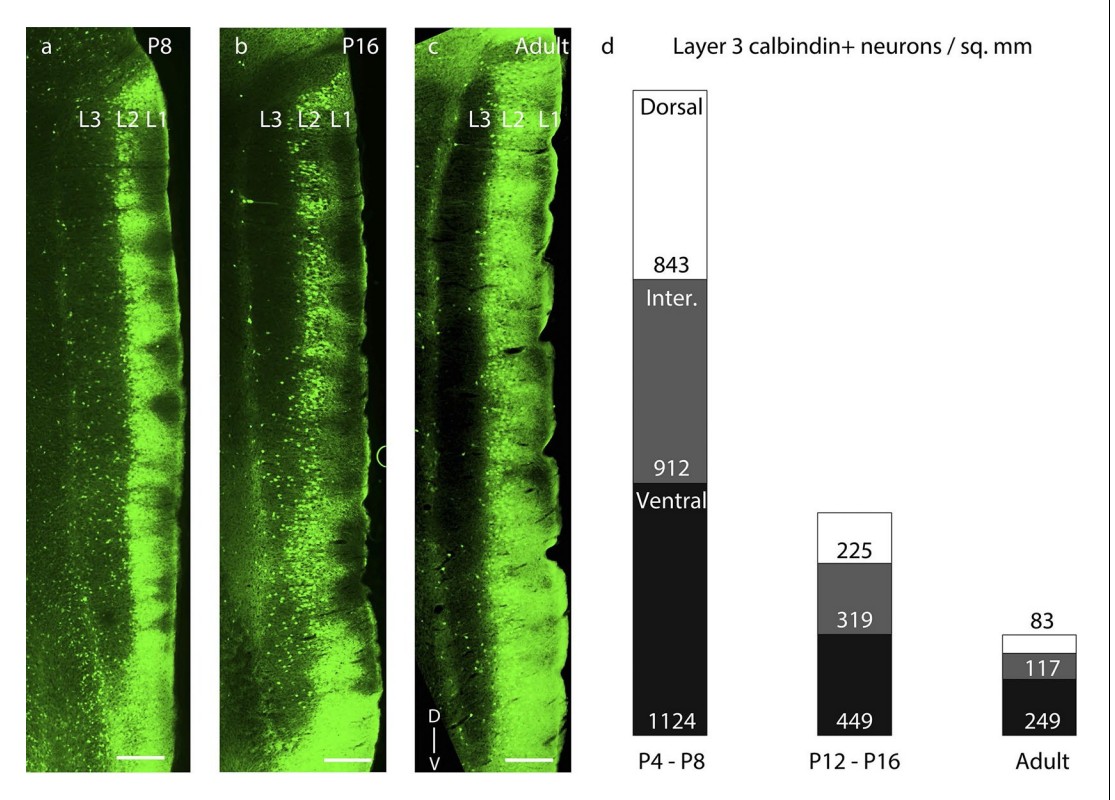

**Figure 5.** Dorsal-to-ventral disappearance of layer 3 calbindin expression. Parasaggital sections showing superficial layers of the MEC processed for calbindin-immunoreactivity (green). (a) Calbindin expression is seen throughout layer 3 in P8 rats. (b) Calbindin expression is seen only in ventral half of layer 3 in P16 rats. (c) Calbindin expression is largely absent in layer 3 in adult rats. (d) Proportion of layer 3 calbindin+ neurons in dorsal (white), intermediate (gray) and ventral (black) MEC in P4-P8 (n=3776 neurons, 8 rats); P12-P16 (n =2014 neurons, 8 rats); and adult (n=828 neurons, 7 rats) rats. The numbers represent layer 3 calbindin+ neuronal density and decay in a dorsal to ventral gradient with age as evident with the reduced proportions of the white (dorsal MEC) and gray (intermediate MEC) sections of the columns with increasing age. Scale bars 250 µm. L1- Layer 1; L2- Layer 2; L3- Layer 3; D- Dorsal; V-Ventral. Orientation in (c) applies to all sections.

The following source data and figure supplements are available for figure 5:

**Figure supplement 1.** Dorsal- ventral distribution of layer 3 reelin expression.

**Figure supplement 1—source data 1.** Calbindin+ neurons (*Figure 5*) and reelin+ neurons (*Figure 5—figure supplement 1*) counted and areas (in µm²) in dorsal, intermediate and ventral parts of layer 3 for determining calbindin+ and reelin+ neuronal densities respectively in P4-P8, P12-P16 and adult rats.

Third, wolframin expression, a marker which co-localizes with calbindin+ pyramidal neurons in layer 2 of MEC in adult rodents (*Kitamura et al., 2014*), develops from dorsal to ventral in layer 2 medial entorhinal cortex and parasubiculum (*Figure 8*). Specifically, wolframin expression starts to appear in the dorsal MEC and the dorsal PaS shortly after birth (*Figure 8a*) and is present only in the dorsal ~10% of the PaS. It extends progressively more ventrally (*Figure 8b*) and covers ~40% at P8 and ~75% at P12 of PaS. At P20 it is expressed throughout the full extent of medial entorhinal cortex and the parasubiculum (*Figure 8c*).

## Discussion

Neurogenesis in the medial entorhinal cortex is completed prior to E18 (*Bayer, 1980a*; *1980b*), and at this time the basic laminar organization of medial entorhinal cortex is already evident. While the basic structure of medial entorhinal cortex appears early, we observe massive developmental

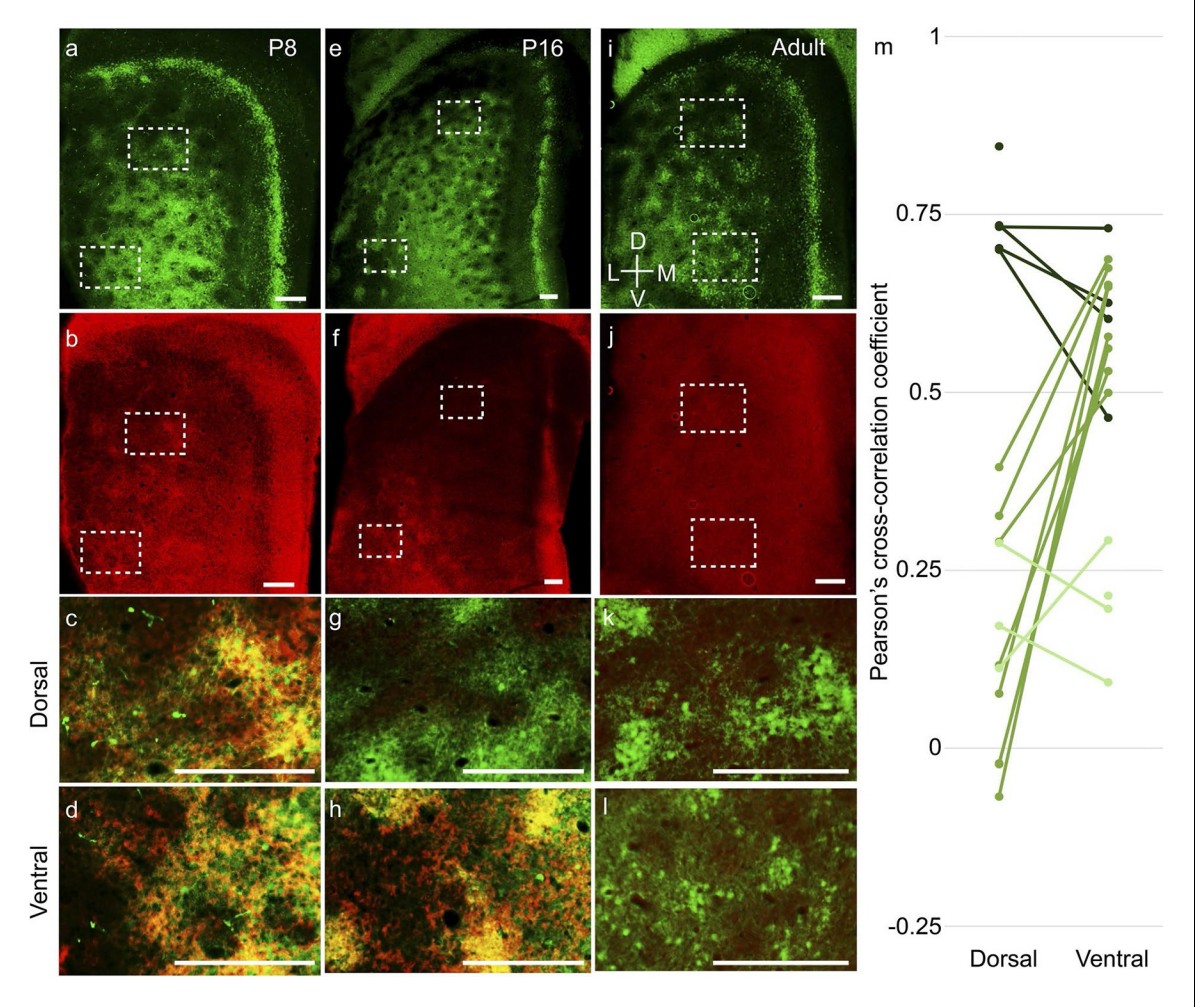

**Figure 6.** Dorsal-to-ventral maturation of layer 2 calbindin+ patches and parasubiculum. Tangential sections of the MEC double-stained for calbindin immunoreactivity (green) and doublecortin immunoreactivity (red). Doublecortin is a marker for immature neurons and disappears in a dorsal-ventral gradient. (a) Calbindin-expression (green) in P8 rats. (b) Doublecortin-expression (red) in P8 rats. Note the presence of doublecortin throughout the dorso-ventral extent of MEC and parasubiculum. (c) Overlay of the dorsal inset region (dashed) in (a) and (b), showing overlap of calbindin and doublecortin (hence the yellowish color). (d) Overlay of the ventral inset region (dashed) in (a) and (b), showing overlap of calbindin and doublecortin. (e–h) as (a–d) for P16 rats, respectively. However, note that dorsal inset region lacks doublecortin (g) while ventral inset region shows overlap of calbindin and doublecortin (h). Also, note the absence of doublecortin in the dorsal but not the ventral parasubiculum (f). (i–k) as (a–d) for adult rats. No doublecortin is present in either dorsal (k) or ventral (l) regions. (m) Spatial cross-correlations of calbindin and doublecortin in MEC showing high overlap in both dorsal and ventral regions in P8-P12 rats (dark green; n=9 regions, 5 rats); low correlation in dorsal but high overlap in ventral in P16-P20 rats (green; n=16 regions, 8 rats) and low correlations in both dorsal and ventral in adult rats (light green; n=7 regions, 4 rats). The Pearson's cross-correlation coefficient can vary from -1 (anti-correlated) through 0 (un-correlated) to 1 (correlated). Scale bars 250 μm. D- Dorsal; V- Ventral; M- Medial; L- Lateral. Orientation in (i) applies to all sections.

changes in the cortical structure, including a doubling of the thickness of the superficial layers during the first postnatal week.

The clustering of layer 2 MEC calbindin+ neurons into patches is also an early developmental event, and key aspects of the grid-layout of calbindin+ neurons are already present at birth. This observation indicates that the periodic structure of patches is a result of genetic signaling rather than spatial experience. Periodic patterns are ubiquitous in nature, and several chemical patterning systems have been explained on the basis of interaction between dynamical systems (*Turing, 1952*). Since it has been suggested that the grid layout of calbindin+ neurons is functionally relevant for grid cell activity (*Brecht, 2014*), it would be interesting to investigate, whether genetic manipulations would result in changes of layout periodicity and have functional effects. The dendritic

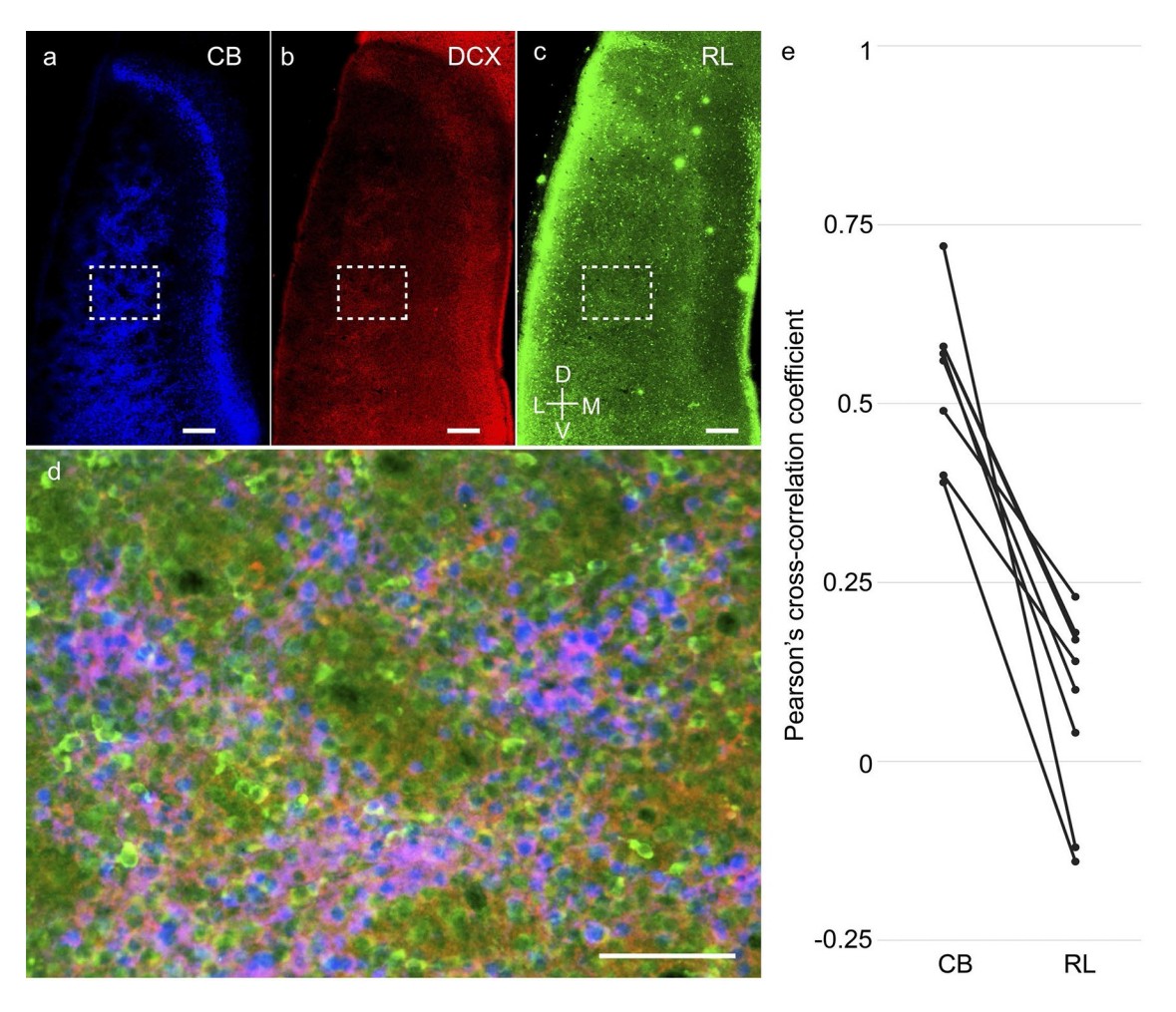

**Figure 7.** Higher co-localization of doublecortin with calbindin+ pyramidal than reelin+ stellate cells in the developing medial entorhinal cortex. Tangential sections of the MEC layer 2 triple-stained for calbindin immunoreactivity (CB; blue), doublecortin immunoreactivity (DCX; red) and reelin immunoreactivity (RL; green). Pyramidal but not stellate cells are structurally immature during early postnatal stages. (a) Calbindin-expression (blue) in layer 2 of MEC. (b) Doublecortin-expression (red) in layer 2 of MEC. (c) Reelin-expression (green) in layer 2 of MEC. (d) Overlay of the inset region (dashed) in (a), (b) and (c), showing a higher co-localization of doublecortin (red) with calbindin (blue), than reelin (green). (e) Spatial cross-correlations of doublecortin with calbindin and reelin showing high overlap of doublecortin with calbindin but not reelin (n=8 regions, 8 rats). Scale bars (a–c) 250 μm; (d) 100 μm. D- Dorsal; V- Ventral; M- Medial; L- Lateral. Orientation in (c) applies to all sections.

clustering of calbindin+ pyramidal neurons is similar to dendritic development in the neocortex (*Petit et al., 1988*) and is established by the end of the first postnatal week. The cholinergic innervation of the calbindin+ patches was present by P4 in line with other long-range connectivity patterns in the MEC (*O'Reilly et al., 2015*), which are also established early in development.

Reelin is an important protein in cortical layer development (*D'Arcangelo et al., 1995*) and in the early stages of postnatal development we see the strongest reelin expression in layer 1, where reelin secreting Cajal-Retzius cells are involved in radial neuronal migration (*Pesold et al., 1998*). Stellate cells in layer 2 of MEC, which can be visualized by reelin-immunoreactivity (*Varga et al., 2010*), were scattered (*Ray et al., 2014*) throughout postnatal development.

Layer 3 of the MEC features a complementary transition of calbindin+ and reelin+ neurons during the first couple of postnatal weeks. While the density of reelin+ neurons increases, there is a concurrent decline in calbindin+ neuronal density in layer 3 of MEC, though part of the calbindin+ neuronal density decline can be attributed to the increasing brain size. Taken together with the presence of radial neuronal migration promoting Cajal-Retzius cells in layer 1 during this period, it would be

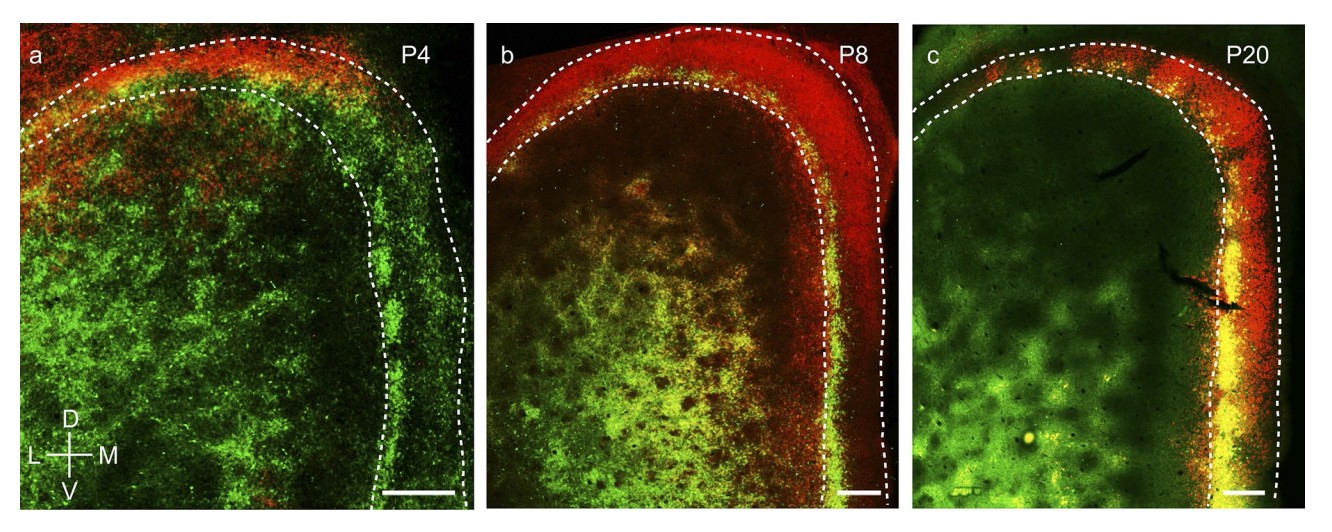

**Figure 8.** Dorsal-to-ventral maturation of wolframin expression in the medial entorhinal cortex and parasubiculum. (a) Tangential sections of the MEC and PaS (outlines dashed) double-stained for calbindin-immunoreactivity (green) and wolframin immunoreactivity (red) in a P4 rat. Shown is an overlay of red and green fluorescence. (b) as (a) for a P8 rat. (c) as (a) for a P20 rat. Wolframin is present in the dorsal ~10% of the parasubiculum at P4, ~40% at P8 and 100% at P20. Note that wolframin expression co-localizes with calbindin-expression in the MEC (hence the yellowish color) and increases from dorsal to ventral with age. Scale bars 250 µm. D- Dorsal; V- Ventral; M- Medial; L- Lateral. Orientation in (a) applies to all sections.

interesting to investigate whether the transient calbindin+ neurons are migrating to layer 2 or changing their phenotype to reelin+ neurons, and what layer and cell-type specific functional differences are observed in this early postnatal development stage.

An interesting observation is the presence of clusters of neurons in the parasubiculum, which transiently express calbindin in early postnatal stages, and subsequently express wolframin. Transient expression of calbindin has been observed in early postnatal development in the neocortex (*Hogan and Berman, 1993*) and midbrain regions (*Liu and Graybiel, 1992*), but its functional significance remains largely unknown. Our data show, however, that at early developmental stages the parasubiculum and medial entorhinal cortex share a similar organization in calbindin+ patches. Additionally, the expression of wolframin in the parasubiculum persists in adults, while calbindin+ neurons in MEC layer 2 also exhibit wolframin (*Kitamura et al., 2014*) from the end of the first postnatal week. Current studies generally focus on cell-type specific investigations using proteins expressed by these cells. However, investigations to study the specific roles of these proteins (*Li et al., 1995*) might provide interesting insights towards understanding the finer differences in the functionalities exhibited by these cells. For instance, calbindin is a calcium buffer, and reduces the concentration of intracellular calcium (*Mattson et al., 1991*), while wolframin is implicated in increasing intracellular calcium levels (*Osman et al., 2003*). With the medial entorhinal cortex and parasubiculum having many similarities in their spatial discharge properties (*Tang et al., 2014*; *Boccara et al., 2010*; *Tang et al., 2016*), a structure-function comparison of the wolframin+/transiently-calbindin+ neurons in the parasubiculum and the wolframin+/ permanently-calbindin+ neurons in the medial entorhinal cortex would be worthwhile.

A dorsal-to-ventral development profile was observed in the superficial layers of the MEC and parasubiculum. This conclusion was suggested by the progressive disappearance of the calbindin expression in layer 3 from dorsal to ventral; the progressive disappearance of doublecortin expression in layer 2 and parasubiculum from dorsal to ventral; and the progressive appearance of the wolframin expression in superficial layer 2 of MEC and parasubiculum from dorsal to ventral. Homing behavior in rats, as well as spontaneous exploratory behavior develops around the end of second postnatal week (*Wills et al., 2014*; *Bulut and Altman, 1974*) while spontaneous exploration of larger environments outside the nest emerge towards the end of the third postnatal week (*Wills et al., 2014*). This is coincident with the timeline of maturation of calbindin+ patches in the dorsal and ventral MEC respectively. Since the dorsal MEC represents smaller spatial scales and the ventral MEC

progressively larger scales (*Hafting et al., 2005*; *Stensola et al., 2012*), these data may indicate that the rat's navigational system matures from small to large scales. Early eyelid opening experiments have indicated an accelerated development of spatial exploratory behaviour (*Kenny and Turkewitz, 1986*; *Foreman and Altaha, 1991*), and similar experiments might provide insights into whether early behavioral development is accompanied by an accelerated development of the microcircuit underlying spatial navigation.

The higher co-localization of doublecortin with calbindin+ pyramidal cells than reelin+ stellate cells, supports further the dichotomy of structure-function relationships exhibited by these two cell types (*Ray et al., 2014*; *Tang et al., 2014*). Grid and border cells have been implicated to be largely specific to pyramidal and stellate cells respectively (*Tang et al., 2014*)and the delayed structural maturation of pyramidal cells might reflect the delayed functional maturation of grid cells (*Wills et al., 2010*; *Langston et al., 2010*), with the converse being applicable to stellate and border cells (*Bjerknes et al., 2014*). The divergent projection patterns of pyramidal and stellate cells, with the former projecting to CA1 (*Kitamura et al., 2014*) and contralateral MEC (*Varga et al., 2010*) and the latter to dentate gyrus (*Varga et al., 2010*; *Ray et al., 2014*) and deep layers of MEC (*Sürmeli et al., 2015*), have differing theoretical interpretations in spatial information processing.

The same sets of neurons, which correspond to grid and border cells (*Tang et al., 2014*), have also been implicated to be differentially involved in temporal association memory (*Kitamura et al., 2014*) and contextual memory (*Kitamura et al., 2015*) respectively. An underlying differential structural maturation timeline of the microcircuit governing these processes may also translate into a differential functional maturation profile of these memories.

We conclude that the structural maturation of medial entorhinal cortex can be coarsely divided into an early appearance of the calbindin+ neuron patches and a progressive cell-type specific refinement of the cellular structure, which proceeds along the dorsal to ventral axis.

## Materials and methods

All experimental procedures were performed according to the German guidelines on animal welfare under the supervision of local ethics committees (LaGeSo) under the permit T0106-14.

### Brain tissue preparation

Male and female Wistar rats (n=83) from E18 to P24 and adults (>P42) were used in the study. The ages were accurate to ± 1 day. Animals were anaesthetized by isoflurane, and then euthanized by an intraperitoneal injection of 20% urethane. They were then perfused transcardially with first 0.9% phosphate buffered saline solution, followed by 4% formaldehyde, from paraformaldehyde, in 0.1 M phosphate buffer (PFA). For prenatal animals, pregnant rats at E18 were perfused in the aforesaid manner and the E18 animals were then extracted from the uterus. Subsequently, brains were removed from the skull and postfixed in PFA overnight. Brains were then transferred to 10% sucrose solution for one night and subsequently immersed in 30% sucrose solution for at least one night for cryoprotection. The brains were embedded in Jung Tissue Freezing Medium (Leica Microsystems Nussloch, Germany), and subsequently mounted on the freezing microtome (Leica 2035 Biocut) to obtain 60 μm thick sagittal sections or tangential sections parallel to the pia.

Tangential sections of the medial entorhinal cortex were obtained by separating the entorhinal cortex from the remaining hemisphere by a cut parallel to the surface of the medial entorhinal cortex (*Video 1*). For subsequent sectioning the surface of the entorhinal cortex was attached to the block face of the microtome.

### Histochemistry and immunohistochemistry

Acetylcholinesterase (AChE) activity was visualized according to previously published procedures (*Ichinohe et al., 2008*; *Tsuji, 1998*). After washing brain sections in a solution containing 1 ml of 0.1 M citrate buffer (pH 6.2) and 9 ml 0.9% NaCl saline solution (CS), sections were incubated with CS containing 3 mM $CuSO_4$, 0.5 mM $K_3Fe(CN)_6$, and 1.8 mM acetylthiocholine iodide for 30 min. After rinsing in PB, reaction products were visualized by incubating the sections in PB containing 0.05% 3,3'- Diaminobenzidine (DAB) and 0.03% nickel ammonium sulfate.

Immunohistochemical stainings were performed according to standard procedures. Briefly, brain sections were pre-incubated in a blocking solution containing 0.1 M PBS, 2% Bovine Serum Albumin

(BSA) and 0.5% Triton X-100 (PBS-X) for an hour at room temperature (RT). Following this, primary antibodies were diluted in a solution containing PBS-X and 1% BSA. Primary antibodies against the calcium binding protein Calbindin (Swant: CB300, CB 38; 1:5000), the extracellular matrix protein Reelin (Millipore: MAB5364; 1:1000), the transmembrane protein Wolframin (Proteintech: 11558-1-AP; 1:200), the microtubule associated protein Doublecortin (Santa Cruz Biotechnology: sc-8086; 1:200) and the calmodulin binding protein Purkinje cell protein 4 (Sigma: HPA005792; 1:200) were used. Incubations with primary antibodies were allowed to proceed for at least 24 hr under mild shaking at 4°C in free-floating sections. Incubations with primary antibodies were followed by detection with secondary antibodies coupled to different fluorophores (Alexa 488, 546 and 633; Invitrogen). Secondary antibodies were diluted (1:500) in PBS-X and the reaction allowed to proceed for two hours in the dark at RT. For multiple antibody labeling, antibodies raised in different host species were used. For visualizing cell nuclei, sections were counterstained with DAPI (Molecular Probes: R37606). After the staining procedure, sections were mounted on gelatin coated glass slides with Vectashield mounting medium (Vectorlabs: H-1000).

## Image acquisition

An Olympus BX51 microscope (Olympus, Shinjuku Tokyo, Japan) equipped with a motorized stage (LUDL Electronics, Hawthorne NY) and a z-encoder (Heidenhain, Shaumburg IL, USA), was used for bright field microscopy. Images were captured using a MBF CX9000 (Optronics, Goleta CA) camera using Neurolucida or StereoInvestigator (MBF Bioscience, Williston VT, USA). A Leica DM5500B epi-fluorescence microscope with a Leica DFC345 FX camera (Leica Microsystems, Mannheim, Germany) was used to image the immunofluorescent sections. Alexa fluorophores were excited using the appropriate filters (Alexa 350 – A4, Alexa 488 – L5, Alexa 546 – N3, Alexa 633 – Y5). Fluorescent images were acquired in monochrome, and color maps were applied to the images post acquisition. Post hoc linear brightness and contrast adjustment were applied uniformly to the image under analysis.

## Analysis of layer width

To determine the width of different layers of the medial entorhinal cortex, we prepared parasagittal sections and stained them for calbindin-immunoreactivity, Purkinje cell protein-immunoreactivity and DAPI. Measurements were taken from dorsal, medial and ventral parts of each section analyzed using Leica Application Suite AF (Leica Microsystems, Mannheim, Germany).

## Analysis of spatial periodicity

To determine the spatial periodicity of calbindin$^+$ patches, we determined spatial autocorrelations. The spatial autocorrelogram was based on Pearson's product moment correlation coefficient (*Sargolini et al., 2006*).

$$r(\tau_x, \tau_y) = \frac{n \sum f(x,y) f(x-\tau_x, y-\tau_y) - \sum f(x,y) \sum f(x-\tau_x, y-\tau_y)}{\sqrt{n \sum f(x,y)^2 - (\sum f(x,y))^2} \sqrt{n \sum f(x-\tau_x, y-\tau_y)^2 - (\sum f(x-\tau_x, y-\tau_y))^2}}$$

where, $r(\tau_x, \tau_y)$ is the autocorrelation between pixels or bins with spatial offset $\tau_x$ and $\tau_y$. $f$ is the monochromatic image without smoothing, n is the number of overlapping pixels. Autocorrelations were not estimated for lags of $\tau_x$ and $\tau_y$, where n<20. Grid scores were calculated, as previously described (*Ray et al., 2014*), and can vary from −2 to 2.

## Analysis of spatial overlap

To determine the degree of overlap between doublecortin and calbindin or reelin, we determined spatial crosscorrelations. Spatial crosscorrelations were determined based on Pearson's product moment correlation coefficient.

$$r = \frac{n \sum f1(x,y) f2(x,y) - \sum f1(x,y) \sum f2(x,y)}{\sqrt{n \sum f1(x,y)^2 - (\sum f1(x,y))^2} \sqrt{n \sum f2(x,y)^2 - (\sum f2(x,y))^2}}$$

where, $r$ is the cross-correlation between the monochromatic images $f1$ and $f2$ without smoothing. n is the number of pixels in the image. The Pearson's cross-correlation coefficient can vary from -1 (anti-correlated) through 0 (un-correlated) to 1 (correlated).

For analysis of dorso-ventral variation in overlap between doublecortin with calbindin, two regions of the same size were selected from a section double-stained for calbindin and doublecortin. One region was selected from the dorsal half of the section and another from the ventral half and the regions were represented as pairs. Where, due to section damage, it was not possible to obtain regions from both dorsal and ventral parts, the data was presented as unpaired.

For analysis of variation in overlap between doublecortin and calbindin/reelin, comparisons were performed between the same regions from a section triple stained for calbindin, reelin and doublecortin.

## Acknowledgements

We thank Juliane Steger and Undine Schneeweiß for outstanding technical support, Shimpei Ishiyama for excellent graphic design and Peter Bennett, Edith Chorev, Andreea Neukirchner, Juan Ignacio Sanguinetti, Jean Simonnet and Robert Naumann for comments.

## Additional information

### Funding

| Funder | Grant reference number | Author |
| --- | --- | --- |
| Bundesministerium für Bildung und Forschung | 01GQ1001A | Michael Brecht |
| European Research Council | | Michael Brecht |
| Gottfried Wilhem Leibniz Prize | | Michael Brecht |
| Deutsche Forschungsgemeinschaft | | Michael Brecht |
| NeuroCure | | Michael Brecht |
| Humboldt-Universität zu Berlin | | Saikat Ray Michael Brecht |
| Bernstein Center for Computational Neuroscience Berlin | | Saikat Ray Michael Brecht |

The funders had no role in study design, data collection and interpretation, or the decision to submit the work for publication.

### Author contributions

SR, Designed the experiments, Wrote the manuscript, Performed the experiments and analyzed the results.; MB, Designed the experiments, Wrote the manuscript, Supervised the project.

### Author ORCIDs

Saikat Ray, http://orcid.org/0000-0002-2732-6612

### Ethics

Animal experimentation: All experimental procedures were performed according to the German guidelines on animal welfare under the supervision of local ethics committees (LaGeSo) under the permit T0106 - 14.

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
