## [Decision Letter]

Thank you for submitting your work entitled "Structural Development and
Dorsoventral Maturation of the Medial Entorhinal Cortex" for consideration by
*eLife*. Your article has been reviewed by three peer reviewers, and
the evaluation has been overseen by Howard Eichenbaum as the Reviewing Editor and Eve
Marder as the Senior Editor.

The following individuals involved in review of your submission have agreed to reveal
their identity: Thomas Van Groen, Andre Fenton, and Rosamund Langston (peer
reviewers).

The reviewers have discussed the reviews with one another and the Reviewing Editor has
drafted this decision to help you prepare a revised submission.

Summary:

This paper provides important new insights into the developmental circuitry of the
medial entorhinal and parasubiculum circuitry, of high importance to considerable
current research on the functional organization of these brain areas.

Essential revisions:

The reviewers had several general concerns about explanation of the findings and many
recommendations for improvement of the data presentation. These are detailed below.

*Reviewer #1:*

The manuscript by Ray and Brecht is quite interesting and relatively well written. The
Introduction, Methods and Discussion are fine, even if the Discussion is a bit short.
The Methods are good, but could be more detailed. The Results, however, need to be
substantially revised or improved. One minor issue is that "superficial
layers" need to be defined, even if most people know those are layers I to III,
another issue are the tangential sections, that should be more clearly explained in the
text. If a cortical area changes in size, by definition neuronal density decreases, this
needs to be more clearly stated and discussed. The biggest issue is the presentation of
the figures in the Results. Figure 1 shows
changes in the size of the superficial layers, but also includes the deep layers. Figure 2: the autocorrelation images need to be the
same size as the insets of the immunohistochemical staining images to be acceptable and
interpretable. The figure orientation is needed in Figure 2; what is where? For instance, what is the band of bright staining
on the right side of the image? The acetylcholinesterase activity staining is of low
quality (at least in the images provided). Similarly, in Figure 3, images showing the sections at equal magnification would be helpful
(orientation?). Figure 5 is of too low quality to
be interpretable, doublecortin is supposed to stain neurons, which is not very visible
in this figure. Taken together, this will make the data more understandable for readers;
in the current version the images are not exceedingly helpful.

*Reviewer #2:*

General assessment:

This is a clear and valuable study, describing the anatomical and developmental
expression of calbindin and other molecular markers of pyramidal cells in the MEC and
parasubiculum across developmental time from late gestation to young adulthood. These
cells are important elements of the microcircuitry that generates spatially-tuned
discharge. Whether the histochemically identified Calbindin^+^ cells correspond
to functionally defined grid cells and/or border cells is controversial, which makes the
present study an important contribution to the debate. The debate is crucial to notions
of how the hippocampal-entorhinal system generates and computes information about space
from the different functional components that have been identified such as distances,
locations, directions, borders and speed. The development of the functional cell classes
has been studied in recent years to constrain the theories, but which
functionally-identified cells correspond to which histochemical and morphological cell
classes is still unclear and clarification of this issue will provide important
knowledge of the circuit wiring diagram that will drive theories of neural computation
that are grounded in structure-function relationships. In summary, this manuscript is an
important contribution.

Summary of concerns:

The manuscript is clearly written and the studies appear to be done carefully, and the
procedures are straightforward. The data look clear and the interpretations of the
measurements are not controversial. There are however a number of improvements I will
suggest that will make the report more accessible to the general readership of
*eLife*.

1) The analyses and interpretation of the histochemical images depend strongly on
knowledge of the anatomical topography of the region, which non-specialists will not be.
Whether the sections are tangential or parasaggital, and knowledge of the precise
cutting angles is important even for the specialist. Consequently, it would be valuable
to provide a 3-D model of the brain or just the cortical region with indications of the
tangential and parasaggital planes and to use these on the Figures as a short-hand to
help orient the reader. I know this is asking too much, but I will mention it to make my
point: a 3-D CLARITY image of the immunolabeled cells would go a long way towards
showing this very cool grid-like organization to the non-specialist and specialist
alike. A 3-D model could accomplish the same.

2) I was disappointed not to see analysis of the stellate ocean cells, the coexistence
of which in the region, but outside the pyramidal patches, is the source of the
controversy. The authors should explain in the manuscript why parallel analyses were not
performed.

3) I like the Discussion, which is appropriately driven by the findings. Again, for the
general readership of *eLife*, I suggest the Discussion be expanded to
more explicitly include the differential hippocampal and neocortical connectivity of the
stellate and pyramidal cells, a discussion of what is known about their function
properties, and why it is important to understanding how information about space is
computed.

*Reviewer #3:*

Although structural maturation has been shown for the hippocampus this was lacking in
the literature for entorhinal cortex despite the huge recent interest in the spatially
selective cell types found there, so this paper is timely and of high theoretical
importance.

The Introduction focuses on Layer II of MEC and is very brief – a little more attention
to the cell types of Layer III and PaS and their electrophysiological characteristics,
and what makes them different to the Layer II neurons would be nice since quite a lot of
the results focus on these and not on layer II.

Also how does one identify stellate neurons and their developmental profile, and why is
this not done? Also why are deeper layers of MEC not analysed? These are not suggestions
for more experiments, just requests to explain briefly why they were not included
here.

A bit more information about wolframin would be good – the authors initially state it is
co-localised with calbindin but from their results this is not always the case?

The figures are mainly well explained, and the layout of the individual images, with the
data presented graphically, is intuitive and aesthetically pleasing. The images clearly
depict what is described and the mean data that is shown in the graphs.

Figure titles and/or images should include whether they depict MEC or PaS (e.g. not
clear from Figure 5 where it is referred to in
the text as being MEC layer II AND PaS as far as I read.)

Discuss relevance of wolframin vs. calbindin and why they appear separately in PaS but
are co-localised in pyramidal cells of MEC layer 2.

In the Discussion it is a very interesting theory that the dorso-ventral structural
development profile may reflect the maturation of the range of the spatial navigation
system however this would be quite difficult to study since coincident with this is the
fact that rats explore further after their eyes open at the end of the second postnatal
week and it would be difficult to manipulate this behaviour to occur earlier, although
similar things have been achieved with experiments to open rat pups' eyelids
earlier.

The following theory suggested in the fourth paragraph of the Discussion that the patchy
doublecortin co-localised with calbindin may reflect different functional maturation of
border and grid cells requires a little more explanation, I did not see any obvious
link.

Overall the paper provides much-needed high-quality solid anatomical evidence for
dorso-ventral structural maturation occurring in the MEC and PaS, similar to that seen
in the hippocampus. This data should feed into various models of neuronal spatial
representations and presents the possibility that subsets of grid cells may mature
before others so maps could potentially be formed of smaller spaces before larger ones,
and a full complement of grid cells may not be necessary to see grid-like firing, a
hypothesis which is testable and intriguing.

---

## [Author Response]

*Essential revisions:*

*The reviewers had several general concerns about explanation of the findings and
many recommendations for improvement of the data presentation. These are detailed
below.* Reviewer #1:

*The manuscript by Ray and Brecht is quite interesting and relatively well
written. The Introduction, Methods and Discussion are fine, even if the Discussion is
a bit short. The Methods are good, but could be more detailed.*

We thank the reviewer for appreciating the manuscript. We agree with the reviewer’s
assessment about the brevity of certain sections of the manuscript and have performed a
significant revision to all sections of the manuscript to result in a more detailed and
comprehensive revised manuscript.

*The Results, however, need to be substantially revised or improved. One minor
issue is that "superficial layers" need to be defined, even if most people
know those are layers I to III, another issue are the tangential sections, that
should be more clearly explained in the text.*

We agree with the reviewer’s assessment and have implemented their suggestion.

Changes:

Results, first paragraph *–* We have addressed the definition of
superficial layer.

Video 1 – We have added a novel schematic
video that illustrates the tangential sectioning process and provides the reader with a
clear perspective of how the tangential sections are obtained from the rodent brain, and
how the structures seen in tangential sections relate to the brain in situ.

*If a cortical area changes in size, by definition neuronal density decreases,
this needs to be more clearly stated and discussed. The biggest issue is the
presentation of the figures in the Results. Figure 1 shows changes in the size of the superficial layers, but also includes
the deep layers.*

We agree with this criticism and thank the reviewer for the suggestion to include
variation in depth in deep layers of the MEC with increasing age.

Changes:

Discussion, fourth paragraph *–* We discuss the effect of increasing
brain size on neuronal density.

Figure 1 and Results, first paragraph
*–* We have updated Figure 1
and the Results to include changes in size in the deep layers of MEC.

*Figure 2: the autocorrelation images need
to be the same size as the insets of the immunohistochemical staining images to be
acceptable and interpretable. The figure orientation is needed in Figure 2; what is where? For instance, what is
the band of bright staining on the right side of the image?*

We thank the reviewer for pointing out this oversight.

Changes:

Figure 2
*–* We have ensured that the inset of the immunohistochemical stain and
the autocorrelation images are of the same size. The scale bar in the autocorrelation
image is half the length of the one in the immunohistochemical image, since the
autocorrelation denotes twice the length and breadth of the immunohistochemical
image.

Figure legends (all relevant figures) *–* We have denoted the orientation
in one panel of every figure, and now we state in the figure legends that it applies to
all the other panels unless explicitly noted.

Figure legend (Figure 2) – We mention in the
figure legend that the bright green band pointed out by the reviewer denotes the
parasubiculum.

*The acetylcholinesterase activity staining is of low quality (at least in the
images provided). Similarly, in Figure 3,
images showing the sections at equal magnification would be helpful
(orientation?).*

We agree with the reviewer’s criticism and have changed the layout of the figure to
address the issue.

Changes:

Figure 2 – We have changed the layout of the
figure to enable better visualization and co-localization of acetylcholinesterase
activity and calbindin immunoreactivity.

Figure legends (all relevant figures) *–* In Figure 3 and elsewhere, as noted above, we have updated the legend
to state that the orientation illustrated in one panel applies to the entire figure
unless explicitly stated.

*Figure 5 is of too low quality to be
interpretable, doublecortin is supposed to stain neurons, which is not very visible
in this figure. Taken together, this will make the data more understandable for
readers; in the current version the images are not exceedingly helpful.*We
agree with the referee’s criticism, about the need for a higher magnification image to
visualize doublecortin localization in a single neuron. However, since other reviewers
found the layout of these images quite intuitive, we decided not to change Figure 5 (now Figure 6) and instead added a novel Figure 7
to address this issue.

Changes:

Figure 7 – We have now added a novel Figure 7, as a result of our new experiments, which
shows at higher magnification the co-localization of doublecortin in the cell membrane
of the cells, with calbindin.

This figure also illustrates reelin which mark stellate cells in layer 2 of MEC, in
response to queries raised by the other reviewers and quantifies the differential
co-localization of doublecortin with calbindin and reelin during early postnatal
development.

Reviewer #2:

*Summary of concerns: The manuscript is clearly written and the studies appear to
be done carefully, and the procedures are straightforward. The data look clear and
the interpretations of the measurements are not controversial. There are however a
number of improvements I will suggest that will make the report more accessible to
the general readership of eLife. 1) The analyses and interpretation of the
histochemical images depend strongly on knowledge of the anatomical topography of the
region, which non-specialists will not be. Whether the sections are tangential or
parasaggital, and knowledge of the precise cutting angles is important even for the
specialist. Consequently, it would be valuable to provide a 3-D model of the brain or
just the cortical region with indications of the tangential and parasaggital planes
and to use these on the Figures as a short-hand to help orient the reader. I know
this is asking too much, but I will mention it to make my point: a 3-D CLARITY image
of the immunolabeled cells would go a long way towards showing this very cool
grid-like organization to the non-specialist and specialist alike. A 3-D model could
accomplish the same.* We thank the reviewer for the idea of a 3D model, and
have included a novel schematic video (Video 1) to address these concerns.

Changes:

Video 1 – We have added this video, which
illustrates the tangential sectioning process and provides the reader with a clear
perspective of how the tangential sections are obtained from the rodent brain, and how
the grid-like structures seen in tangential sections relate to the brain in situ.

*2) I was disappointed not to see analysis of the stellate ocean cells, the
coexistence of which in the region, but outside the pyramidal patches, is the source
of the controversy. The authors should explain in the manuscript why parallel
analyses were not performed.* This comment is similar to the criticism of
another reviewer and we think the omission of stellate cell development was a major flaw
in the previous version of our manuscript. We have performed a series of experiments to
address this concern, and now provide a detailed perspective of the development of
stellate cells.

Changes:

Figure 2—figure supplement 1 – New figure,
illustrating the spatial layout of reelin+ stellate cells during development.

Figure 7 – We have added this new figure,
illustrating the co-localization of doublecortin with pyramidal but not stellate cells
during early postnatal development.

We have also made appropriate changes in the Abstract, Results and Discussion sections
of the manuscript to reflect these novel results.

In addition, we found an increasing expression of reelin in layer 3 neurons with
development, which we have highlighted in new Figure 4 and new Figure 5—figure supplement 1.

*3) I like the Discussion, which is appropriately driven by the findings. Again,
for the general readership of eLife, I suggest the Discussion be expanded to more
explicitly include the differential hippocampal and neocortical connectivity of the
stellate and pyramidal cells, a discussion of what is known about their function
properties, and why it is important to understanding how information about space is
computed.* We thank the reviewer for the appreciation and the idea to
highlight the differential connectivity profiles of pyramidal and stellate cells and its
impact in spatial information processing.

Changes:

Discussion, seventh paragraph – Discussion on different projection patterns of pyramidal
and stellate cells.

Reviewer #3:

*Although structural maturation has been shown for the hippocampus this was
lacking in the literature for entorhinal cortex despite the huge recent interest in
the spatially selective cell types found there, so this paper is timely and of high
theoretical importance. The Introduction focuses on Layer II of MEC and is very brief
– a little more attention to the cell types of Layer III and PaS and their
electrophysiological characteristics, and what makes them different to the Layer II
neurons would be nice since quite a lot of the results focus on these and not on
layer II.* We thank the reviewer for pointing out this oversight, and we have
expanded the Introduction to include the characteristics of cells in layer 3 and
parasubiculum in addition to layer 2.

Changes:

Introduction, fourth paragraph – We have included an introduction on layer 3 and the
functional characteristics of the cells found there.

Introduction, fifth paragraph – We have included an introduction on parasubicuulum and
the functional characteristics of the cells found there.

*Also how does one identify stellate neurons and their developmental profile, and
why is this not done? Also why are deeper layers of MEC not analysed? These are not
suggestions for more experiments, just requests to explain briefly why they were not
included here.* This comment is similar to a query raised by another
reviewer, and we have performed a series of novel experiments to explore the development
of stellates by using the extracellular matrix protein Reelin. We have performed a
series of experiments to address this concern, and now provide a detailed perspective of
the development of stellate cells. For deep layers, we added the development of their
thickness during development, congruent with the recommendations of another
reviewer.

Changes:

Figure 2—figure supplement 1 – New figure,
illustrating the spatial layout of reelin+ stellate cells during development.

Figure 7 – New figure, illustrating the
co-localization of doublecortin with pyramidal but not stellate cells during early
postnatal development.

Figure 1; Results, first paragraph– We have
updated Figure 1 and the Results to include
changes in size in the deep layers of MEC.

We have also made appropriate changes in the Abstract, Results and Discussion sections
of the manuscript to reflect these novel results.

In addition, we found an increasing expression of reelin in layer 3 neurons with
development, which we have highlighted in new Figure 4 and new Figure 5—figure supplement 1.

*A bit more information about wolframin would be good – the authors initially
state it is co-localised with calbindin but from their results this is not always the
case?* We have now expanded our discussion on wolframin on its
co-localization with calbindin in layer 2 of MEC but not the developed
parasubiculum.

Changes:

Discussion, fifth paragraph – We expanded the Discussion to include how studies target
specific cell-types using their molecular profiles, and how experiments towards
understanding the specific roles of these proteins might provide insights in
understanding the functional differences in these cells.

*The figures are mainly well explained, and the layout of the individual images,
with the data presented graphically, is intuitive and aesthetically pleasing. The
images clearly depict what is described and the mean data that is shown in the
graphs.*

We thank the reviewer for the appreciation of the figures.

*Figure titles and/or images should include whether they depict MEC or PaS (e.g.
not clear from Figure 5 where it is referred
to in the text as being MEC layer II AND PaS as far as I read.)* We thank the
reviewer for pointing out this oversight, and have now corrected the figure title and
legend.

*Discuss relevance of wolframin vs. calbindin and why they appear separately in
PaS but are co-localised in pyramidal cells of MEC layer 2.*As noted above,
we provide a more comprehensive discussion on wolframin and calbindin, and their
possible functions.

Changes:

Discussion, fifth paragraph – Discussion on the roles of wolframin and calbindin.

*In the Discussion it is a very interesting theory that the dorso-ventral
structural development profile may reflect the maturation of the range of the spatial
navigation system however this would be quite difficult to study since coincident
with this is the fact that rats explore further after their eyes open at the end of
the second postnatal week and it would be difficult to manipulate this behaviour to
occur earlier, although similar things have been achieved with experiments to open
rat pups' eyelids earlier.* We thank the reviewer for finding the theory
interesting, and have now discussed in greater detail how it can be tested by the early
eyelid opening experiments, as pointed out by the reviewer.

Changes:

Discussion, sixth paragraph – Discussion on possible experiments to test dorso-ventral
maturation hypothesis.

*The following theory suggested in the fourth paragraph of the Discussion that
the patchy doublecortin co-localised with calbindin may reflect different functional
maturation of border and grid cells requires a little more explanation, I did not see
any obvious link.* We apologize for not having adequately clarified this
point, and have now addressed this issue in greater detail by performing a series of
experiments to simultaneously analyze the co-localization of the immature neuronal
marker doublecortin, with pyramidal cell marker calbindin, and stellate cell marker
reelin in layer 2 of MEC. We find a differential neuronal maturation profile, which
indicates that the structural development of pyramidal and stellate cells, closely
mirrors the differential functional maturation profiles of grid and border cells
respectively.

Changes:

Figure 7 – New figure, illustrating the
co-localization of doublecortin with pyramidal and not stellate cells during early
postnatal development.

We have also made appropriate changes in the Abstract, Results and Discussion sections
of the manuscript to reflect these novel results.

*Overall the paper provides much-needed high-quality solid anatomical evidence
for dorso-ventral structural maturation occurring in the MEC and PaS, similar to that
seen in the hippocampus. This data should feed into various models of neuronal
spatial representations and presents the possibility that subsets of grid cells may
mature before others so maps could potentially be formed of smaller spaces before
larger ones, and a full complement of grid cells may not be necessary to see
grid-like firing, a hypothesis which is testable and intriguing.*

We thank the reviewer for the comprehensive summary of our manuscript, which mirrors our
enthusiasm.